# The Microbiological Characteristics and Genomic Surveillance of Carbapenem-Resistant *Klebsiella pneumoniae* Isolated from Clinical Samples

**DOI:** 10.3390/microorganisms13071577

**Published:** 2025-07-04

**Authors:** Mehwish Rizvi, Noman Khan, Ambreen Fatima, Rabia Bushra, Ale Zehra, Farah Saeed, Khitab Gul

**Affiliations:** 1Department of Pharmaceutics, Faculty of Pharmaceutical Sciences, Dow College of Pharmacy, Dow University of Health Sciences, Karachi 75280, Pakistan; rabia.bushra@uok.edu.pk; 2Department of Biosciences, Mohammad Ali Jinnah University, Karachi 75400, Pakistan; knoman915@gmail.com; 3Department of Pathology, Dow International Medical College, Dow University of Health Sciences, Karachi 75280, Pakistan; ambreen.fatima@duhs.edu.pk; 4Department of Pharmacy Practices, Faculty of Pharmaceutical Sciences, Dow College of Pharmacy, Dow University of Health Sciences, Karachi 75280, Pakistan; ale.zehra@duhs.edu.pk; 5Department of Pharmacognosy, Faculty of Pharmaceutical Sciences, Dow University of Health Sciences, Karachi 75280, Pakistan; farah.saeed@duhs.edu.pk

**Keywords:** molecular epidemiology, whole genome sequencing, carbapenem resistant *K. pneumoniae*, antimicrobial resistance

## Abstract

*Klebsiella pneumoniae* is a major public health concern due to its role in Gram-negative bacteremia, which leads to high mortality and increased healthcare costs. This study characterizes phenotypic and genomic features of *K. pneumoniae* isolates from clinical samples in Karachi, Pakistan. Among 507 isolates, 213 (42%) were carbapenem-resistant based on disk diffusion and MIC testing. Urine (29.7%) and blood (28.3%) were the most common sources, with infections predominantly affecting males (64.7%) and individuals aged 50–70 years. Colistin was the only antibiotic showing consistent activity against these isolates. The whole-genome sequencing of 24 carbapenem-resistant *K. pneumoniae* (CR-KP) isolates revealed *bla*_NDM-5_ (45.8%) as the dominant carbapenemase gene, followed by *bla*_NDM-1_ (12.5%) and *bla*_OXA-232_ (54.2%). Other detected *bla*_OXA_ variants included *bla*_OXA-1_, *bla*_OXA-4_, *bla*_OXA-10_, and *bla*_OXA-18_. The predominant beta-lactamase gene was *bla*_CTX-M-15_ (91.6%), followed by *bla*_CTX-M-163_, *bla*_CTX-M-186_, and *bla*_CTX-M-194_. Sequence types ST147, ST231, ST29, and ST11 were associated with resistance. Plasmid profiling revealed IncR (61.5%), IncL (15.4%), and IncC (7.7%) as common plasmid types. Importantly, resistance was driven not only by acquired genes but also by chromosomal mutations. Porin mutations in OmpK36 and OmpK37 (e.g., P170M, I128M, N230G, A217S) reduced drug influx, while *acrR* and *ramR* mutations (e.g., P161R, G164A, P157*) led to efflux pump overexpression, enhancing resistance to fluoroquinolones and tigecycline. These findings highlight a complex resistance landscape driven by diverse carbapenemases and ESBLs, underlining the urgent need for robust antimicrobial stewardship and surveillance strategies.

## 1. Introduction

The escalating prevalence of carbapenem-resistant *Klebsiella pneumoniae* has placed it among the World Health Organization’s (WHO) 2024 priority bacterial pathogens, underscoring its critical impact on public health, particularly in Pakistan [1,2]. The epidemiology of *K. pneumoniae* infections in Pakistan is characterized by a complex interplay of factors, including antibiotic usage patterns, infection control practices, and underlying patient demographics [3]. Due to its adaptability to diverse environments and resistance to multiple drugs, it is a critical concern in modern healthcare settings [4]. Furthermore, patients with compromised immune systems are particularly susceptible to these infections.

Clinical infections caused by this pathogen are often resistant to multiple antibiotics, resulting in mortality rates ranging from 40% to 70% [5,6]. Over the past two decades, *K. pneumoniae* has developed resistance to several critical classes of antibiotics, including beta-lactams, aminoglycosides, fluoroquinolones, tetracyclines, sulfonamides, fosfomycin, and phenicols. Worryingly, its resistance to carbapenems, considered the last line of defense in antibiotic treatment, is growing rapidly [7,8].

Carbapenem resistance in *K. pneumoniae* (CRKP) is primarily attributed to the production of various enzymes, including *Klebsiella pneumonmiae* carbapenemases (KPC), New Delhi metallo-beta-lactamases (NDM), oxacillinases (OXA), Verona integron-encoded metallo-beta-lactamases (VIM), and imipenemases (IMP) [9]. The presence of multiple carbapenemase genes in some isolates, along with their geographical variability, highlights the complexity of tackling this public health challenge [10]. These antimicrobial genes have emerged from different regions across the globe, with a prevalence ranging from 0.13% to 22% and a pooled prevalence of 5.43% [11].

Beta-lactam resistance in *K. pneumoniae* is primarily driven by the production of various beta-lactamase enzymes. These include extended-spectrum beta-lactamases (ESBLs), which confer resistance to penicillins, third-generation cephalosporins, and aztreonam, but are typically inhibited by beta-lactamase inhibitors such as clavulanic acid [12]. Plasmid-mediated AmpC beta-lactamases (p-AmpC) confer resistance to cephamycins and are not inhibited by classical beta-lactamase inhibitors, making them more difficult to treat. These enzymes can also compromise the efficacy of extended-spectrum cephalosporins when overexpressed [13].

Narrow-spectrum beta-lactamases hydrolyze early-generation penicillins and cephalosporins but do not act on extended-spectrum beta-lactams. Importantly, these beta-lactamases are frequently co-located with other resistance determinants, including carbapenemases and genes conferring resistance to aminoglycosides or fluoroquinolones, on mobile genetic elements like plasmids, integrons, or transposons, thereby enhancing their dissemination across bacterial populations and increasing the therapeutic challenge.

In Pakistan, antimicrobial resistance (AMR) poses a major threat to the healthcare system, contributing to 1.27 million fatalities and accounting for 20% of deaths in children under the age of five years [14]. Factors such as the overuse and misuse of antibiotics, inadequate infection control practices, and the horizontal transfer of resistance genes among bacteria have accelerated the emergence and spread of carbapenem resistance [15]. Moreover, insufficient planning, ineffective surveillance, and deficiencies within the healthcare system have exacerbated the severity of AMR in the country.

In response, Pakistan established the National AMR Surveillance System in 2015 to address this issue. However, current data remains inadequate for precise assessment [14]. The World Health Organization (WHO) has emphasized that genomic surveillance of multidrug-resistant (MDR) pathogens is both essential and strongly recommended for the global management of AMR [15]. Given this background, the present study aims to elucidate the current state of antibiotic resistance among *K. pneumoniae* isolates in Karachi, Pakistan, through a comprehensive phenotypic and genotypic evaluation.

## 2. Materials and Methods

### 2.1. Sources and Acquisition of Clinical Isolates

The 6125 clinical isolates came from clinical samples of urine, blood, pus, body fluids, sputum, and tracheal aspirate from the hospitalized patients of Dow university hospital from July 2022 to June 2023 submitted to the microbiology lab, Dow diagnostic reference and research laboratory of Dow University of Health Sciences Ojha campus, Karachi, Pakistan.

### 2.2. Isolation and Identification of Bacterial Isolates

The samples were inoculated on 5% sheep blood agar and MacConkey agar plates and then incubated at 37 °C for 18–24 h standard microbiological standards. The initial identification for *K. pneumoniae* was performed by sugar identification tests, which include Triple Sugar Iron (TSI), citrate, urea, sulfide indole motility (SIM), and finally confirmed by VITEK^®^ card GN (Biomerieux, Marcy-l’Étoile, France).

### 2.3. Antimicrobial Susceptibility Testing (AST)

The Kirby–Bauer disk diffusion test was performed using carbapenem class antibiotics, including imipenem (10 µg), meropenem (10 µg), ertapenem (10 µg), and doripenem (10 µg); a zone of inhibition less than or equal to 18 mm was considered as resistance against ertapenem, while for the other carbapenems, a zone of inhibition less than or equal to 19 mm was taken as resistant. ATCC *Escherichia coli* 25922 was used as the quality control. Resistant strains based on disk diffusion results were further subjected to minimum inhibitory concentration (MIC) testing of CRE by a VITEK N240 card, which comprised carbapenem antibiotics and thirteen other antibiotics, including ticarcillin, ticarcillin–clavulanic acid, piperacillin, piperacillin–tazobactam, ceftazidime, cefepime, aztreonam, amikacin, gentamicin, tobramycin, ciprofloxacin, levofloxacin, and trimethoprim–sulfamethoxazole, as per CLSI guidelines 2023 [16].

### 2.4. Phenotypic Detection of Carbapenemases

For the presence of carbapenemase synthesis in carbapenem-resistant *K. pneumoniae*, Rapidec^®^ carba NP kits were utilized (CRKP), which detect the product formed after the carbapenemase-mediated hydrolysis. The standards for quality control were ATCC *K. pneumoniae* BAA 1706 (carbapenemase negative) and ATCC *K. pneumoniae* BAA 1705 (carbapenemase positive). For whole-genome sequencing (WGS), twenty percent of the *K. pneumoniae* isolates that produced carbapenemase from various clinical specimens with an MIC > 16 µg/mL were chosen at random.

### 2.5. DNA Extraction and Whole-Genome Sequencing

Following screening, 24 CRKP isolates were subjected to genomic DNA isolation. DNA extraction was performed using the cetyltrimethylammonium bromide (CTAB) method. CTAB (Thermo Fisher Scientific, Waltham, MA, USA) is a cationic detergent that facilitates cell lysis and solubilizes cellular membranes, thereby releasing DNA into the solution. The sample is subsequently treated with chloroform (Merck, Darmstadt, Germany) to separate proteins and polysaccharides. Finally, DNA is precipitated using 70% ethanol (Merck) and purified. Genomic DNA was isolated from the overnight-grown culture of the strain. The integrity, purity, and orientation of the genomic DNA at 260/280 nm were evaluated using an Evolution^TM^ Pro UV–Vis Spectrophotometer (Thermo Fisher Scientific). Subsequently, gel electrophoresis (1%) was conducted for 40 min at 90 V.

Genome sequencing was outsourced to BGI-Genomics. For DNA fragment selection, the Agencourt AMPure XP-Medium kit (Brea, CA, USA) was utilized to isolate fragments predominantly ranging from 200 to 400 bp. These fragments underwent end repair, 3′ adenylation, and adapter ligation. Following this, PCR amplification was conducted to obtain adapter-ligated fragments. The AMPure XP-Medium kit was again employed to purify the PCR products. During library preparation, the double-stranded DNA from the PCR products was heat-denatured to produce single-stranded DNA circles (ssCir DNA), which were circularized using a splint oligo sequence.

The prepared libraries underwent sequencing (DNBSEQ^TM^), and quality control was ensured. DNA nanoballs (DNBs) were generated by amplifying the single-stranded circular DNA via a rolling-cycle replication process, resulting in more than 300 copies of each molecule. These DNBs were then loaded onto high-density DNA nanochips. Finally, paired-end 100 bp reads were generated using the combinatorial probe-anchor synthesis (cPAS) technique.

### 2.6. Bioinformatics Analysis of Raw Sequence Data

Following sequencing, FastQC (v. 0.11.9) was employed to evaluate the quality of the resulting FASTQ files. The analysis included assessments of sequence quality, per-base sequence content, sequence length distribution, overrepresented sequences, and adapter content. To ensure data integrity, low-quality reads and overrepresented sequences were removed [17].

The raw sequence data was assembled using SPAdes (v. 3.15.2), and its quality was subsequently assessed with QUAST [18,19]. QUAST performed internal quality checks and provided a range of critical metrics, including the number of contigs, the N50 value (a measure of assembly contiguity), the overall genome size, and the frequency of misassembles. These evaluations allowed for a comprehensive assessment of genome assembly quality.

### 2.7. Multilocus Sequence Typing

Multilocus sequence typing (MLST) was performed using the MLST 2.0 tool (https://cge.food.dtu.dk/services/MLST, accessed on 15 June 2023), which analyzes assembled genome sequences in FASTA format. This tool identifies allelic profiles by detecting seven housekeeping genes—*gapA*, *infB*, *mdh*, *pgi*, *phoE*, *rpoB*, and *tonB*—to determine the sequence type (ST) of each isolate.

### 2.8. Genotypic Evaluation of Antimicrobial Resistance Genes, Plasmids, and Virulence Genes

The assembled genome in FASTA format served as the input file for the genotypic assessment of virulence factors, plasmids, and antibiotic resistance genes (ARGs). The database ResFinder 4.1 (https://cge.food.dtu.dk/services/ResFinder/, accessed on 20 June 2023) was used to detect antimicrobial resistance genes using alignment. The resfinder database was filtered by maintaining sequence coverage at 60% and identity over 90%. Plasmids were found using the Plasmidfinder 2.1 database (https://cge.food.dtu.dk/services/PlasmidFinder/, accessed on 24 June 2023). The coverage was more than 60%, and the minimum percentage identity was more than 95%, in accordance with the default guidelines for plasmid replicon identification. VFDB (https://www.mgc.ac.cn/VFs/, accessed on 27 June 2023) revealed virulence factor genes associated with adhesion, invasion of epithelial cells, intracellular survival, and evading host defense [20,21,22].

### 2.9. Detection of Acquired Resistance and Mutations

The detection of acquired resistance genes and point mutations mediating resistance within the isolates was carried out using the online platform ResFinder 4.1 (https://cge.food.dtu.dk/services/ResFinder/, accessed on 16 August 2023). ResFinder utilizes assembled genomes in FASTA format as input files to identify resistance genes and mutations, providing crucial insights into the genetic basis of antimicrobial resistance in the isolates [23].

### 2.10. Comparative Analysis of Carbapenem Resistance in Pakistan

By identifying the genes responsible for carbapenem resistance, a comparative analysis of carbapenem-resistant *K. pneumoniae* (CR-KP) was conducted. A total of 24 CR-KP isolates were classified through sequence typing as ST147, ST231, ST29, and ST11. The “Pathogenwatch Database” contained 394 isolates, and selections were made based on sequence type and regional representation while applying the exclusion criteria. Including all 24 CR-KP isolates, a total of 82 isolates were gathered between 1998 and 2023. Antimicrobial resistance genotypes and their phenotypic effects were evaluated as part of the comparative analysis. To assess the association between sequence types (STs) and the presence of specific antibiotic resistance genes, a chi-square test of independence was performed. This test evaluates whether the distribution of resistance genes is significantly different across various STs. The data consisted of counts of each resistance gene detected in four predominant STs: ST-147, ST-231, ST-29, and ST-11. Only genes with non-zero counts were included in the analysis.

A phylogenetic tree was constructed to elucidate the genetic relationships among these isolates. Single-nucleotide polymorphism (SNP)-based phylogenetic analysis began with the construction of the core genome using “Snippy” (v. 3.2) [24]. The core.aln file, generated by the “Snippy core” output, was utilized for the subsequent tree construction using the “FastTree” program (v. 2.1.10) [25]. Finally, the trees were visualized using MEGA (v. 11.0.13) software [26]. This comprehensive approach enabled the accurate analysis and visualization of evolutionary relationships among the studied strains.

## 3. Results

During the study period, 6125 isolates were collected in total, out of which 2450 (40.0%) were identified as *Enterobacterales*. Among these, 509 (8.31%) were identified as *K. pneumoniae*. The sample distribution is presented in Figure 1.

Carbapenem resistance was detected in 213 (42%) of the *K. pneumoniae* isolates based on disk diffusion antimicrobial susceptibility testing against carbapenems. Urine (29.7%), blood (28.3%), tracheal aspirate (10.7%), pus (10.2%), and sputum (6.5%) were the most frequently collected sample types for carbapenem-resistant *K. pneumoniae*, with the remaining sample types accounting for less than 4% of the total. Males accounted for 64.7% of these isolates, while females made for 35.3%. The average age (±SD) of the *K. pneumoniae*-infected research participants was 51.4 ± 20.2 years. They ranged in age from 1 year to 97 years. Subsequent analysis revealed that 37.9% of infections occurred in people aged 50 to 70 years. More than one-third (37.5%) of all isolates of carbapenem-resistant *K. pneumoniae* (CR-KP) were recovered from critical care units.

The carbapenem-resistant isolates were further subjected to carba NP testing, and 183 (85%) of the 213 carbapenem-resistant isolates were positive. Minimum inhibitory concentrations (MICs) were then determined, revealing that 119 isolates had an MIC >16 µg/mL against either imipenem or meropenem. Of these, 65 isolates had an MIC >16 µg/mL for both drugs, and 24 (37%) of these isolates were randomly selected for whole-genome sequencing (WGS). The results of Kirby–Bauer disk diffusion antimicrobial susceptibility testing revealed resistance of *K. pneumoniae* against meropenem (10 μg) at 41.92%, imipenem (10 μg) at 42.2%, ertapenem (10 μg) at 42%, and doripenem (10 μg) at 42.4%. Hence, the clinical isolates were identified as carbapenem-resistant *Klebsiella pneumoniae* (CR-KP).

### 3.1. Antibiotic Susceptibillity

The minimum inhibitory concentration (MIC) results of the carbapenem-resistant *K. pneumoniae* (CRKP) isolates against various antibiotics are presented in Table 1. The confirmed resistance of these isolates to multiple antibiotic classes validates their classification as multidrug-resistant (MDR) pathogens, which are characterized by their persistence and the considerable challenges they pose to effective treatment. MIC values of 4 μg/mL or higher indicate that all isolates are resistant to carbapenems, including imipenem (IMP) and meropenem (MEM).

Piperacillin (PIP) and ticarcillin (TIC) are ineffective against these isolates, as they display an MIC of 128 μg/mL. Even in combination with beta-lactamase inhibitors, such as piperacillin–tazobactam (TZP) and ticarcillin–clavulanate (TIM), resistance remains high (128 μg/mL), demonstrating that beta-lactamase inhibitors fail to overcome the resistance mechanisms in these isolates. MIC values of 64 R reveal strong resistance to ceftazidime (CAZ) and cefepime (FEP), suggesting that cephalosporins are ineffective, potentially due to the production of extended-spectrum beta-lactamases (ESBLs) or other resistance mechanisms.

The susceptibility to aminoglycosides, such as gentamicin (CN) and amikacin (AK), varies among isolates. While most isolates, such as KP-56, exhibited significant resistance (64 μg/mL), others remained susceptible at lower doses (2 μg/mL for AK, 1 μg/mL for CN). Most isolates exhibited resistance to tobramycin (TOB), with MIC values of 16 μg/mL, indicating limited therapeutic efficacy of aminoglycoside.

All isolates demonstrate resistance to fluoroquinolones, including ciprofloxacin (CIP) and levofloxacin (LEV), with MIC values ranging from 4 μg/mL to 8 μg/mL. This suggests that fluoroquinolones are not effective for treating infections caused by these strains. Many isolates also exhibit resistance to trimethoprim–sulfamethoxazole (SXT), with MIC values of 320 μg/mL, further narrowing the available treatment options.

Among the tested antibiotics, colistin is the only one showing consistent intermediate sensitivity, with an MIC value of 2 μg/mL across all isolates. Although colistin is commonly used as a last-resort treatment of infection caused by carbapenem-resistant pathogens, its associated toxicity and the potential for resistance emergence pose significant concerns. Additionally, all isolates remain resistant to aztreonam (64 μg/mL). Aztreonam remains effective against Gram-negative bacteria-producing metallo-β-lactamases, as it is not hydrolyzed by these enzymes. However, its efficacy can be compromised by other β-lactamases, like ESBLs and p-AmpC. Therefore, aztreonam, especially when combined with β-lactamase inhibitors targeting ESBLs and AmpC enzymes, offers a promising treatment option for isolates carrying *NDM* genes.

### 3.2. Assembly of Sequence File

The raw sequence files in FASTQ format were uploaded to the NCBI BioProject under ID PRJNA1012456 after sequencing all 24 CR-KP clinical isolates. Following sequencing, the files underwent quality evaluation using FASTQC Version 0.12.0, which confirmed the presence of high-quality matrices. These sequenced files were subsequently utilized for downstream analyses.

The assembled sequences revealed a mean GC content of 56.98% ± 0.15%, a mean N50 value of 197,764 ± 48,726 base pairs, and an average genome size of 5,564,889 ± 130,094 base pairs.

### 3.3. Multilocus Sequence Typing

Among the 24 CR-KPN isolates, multilocus sequencing analysis identified four dominant sequence types. The most prevalent sequence type was ST147 (53.84%), followed by ST231 (30.7%), ST11 (7.6%), and ST29 (7.6%).

### 3.4. Genotypic Profile of Carbapenem-Resistant K. pneumoniae (CRKP) Isolates

The genotypic profile clarifies the presence of several critical resistance genes, including *bla*_CTX-M_, *bla*_TEM_, *bla*_SHV_, *bla*_OXA_, *bla*_NDM_, and *bla*_CMY_, which encode enzymes such as extended-spectrum β-lactamases (ESBLs), carbapenemases, AmpC β-lactamases, and narrow-spectrum β-lactamases, contributing to resistance against a broad range of β-lactam antibiotics.

Among the ESBL genes, four variants of *bla*_CTX-M_ were identified, with *bla*_CTX-M-15_ being the most prevalent (91.6%), followed by *bla*_CTX-M-163_, *bla*_CTX-M-186_, and *bla*_CTX-M-194_, each found in 16.6% of isolates; notably, all four variants were present in two isolates.

The *bla*_TEM-1B_ genotype was detected in 70.8% of isolates, while *bla*_TEM-1A_ was found in 12.5%. These are classified as narrow-spectrum β-lactamase genes, primarily hydrolyzing penicillins. Eight *bla*_SHV_ variants were observed, including *bla*_SHV-11_ (41.7%), *bla*_SHV-67_ (37.5%), *bla*_SHV-28_ (29.1%), *bla*_SHV-106_ (29.1%), *bla*_SHV-182_ (12.5%), *bla*_SHV-187_ (8.3%), *bla*_SHV-12_ (4.1%), and *bla*_SHV-57_ (4.1%).

Carbapenem resistance was predominantly associated with *bla*_OXA_ and *bla*_NDM_ genes. Among the *bla*_OXA_ variants, *bla*_OXA-232_ was the most common (54.2%), followed by *bla*_OXA-1_ (25.0%), *bla*_OXA-48_ (16.6%), *bla*_OXA-181_ (8.3%), and *bla*_OXA-10_ (8.3%). It is important to note that these *bla_OXA_* variants are not ESBL genes but belong to class D β-lactamase genes with carbapenemase activity. The *bla*_NDM-5_ and *bla*_NDM-1_ variants were identified in 45.8% and 12.5% of isolates, respectively. The presence of these *bla*_OXA_ and *bla_NDM_* genes confirm carbapenem resistance through the production of carbapenemases.

Additionally, the plasmid-mediated AmpC β-lactamase gene *bla*_CMY-6_ was detected in 4.1% of isolates, further contributing to the complexity of the resistance mechanisms observed. Since *bla*_CMY-6_ is associated with AmpC β-lactamase production rather than narrow-spectrum β-lactamase activity, cefoxitin resistance should be evaluated to confirm its functional expression. In addition to carbapenem and beta-lactam resistance, the genotypic profile of the isolates also indicated resistance to fluoroquinolones, disinfectants, aminoglycosides, tetracyclines, and phenolics. Figure 2 provides a detailed overview of the genetic markers conferring resistance to various antibiotics.

### 3.5. Virulence Factor Profiling

A comprehensive analysis of virulence factors revealed several genes consistently present across all isolates, highlighting their essential role in pathogenicity. Due to limited clinical metadata, no direct associations could be drawn between virulence factors, resistance genes, and the clinical presentation of infections. Seven classes of virulence factors were universally detected, contributing to biofilm formation, capsule production, and adhesion. All isolates exhibited efflux pumps to expel toxic compounds, while *K. pneumoniae* specifically demonstrated an iron acquisition system. Additionally, the presence of secretion systems, regulatory genes, and serum resistance mechanisms further enhanced overall virulence (Appendix A). Together, these factors enable the pathogen to effectively establish and sustain infections.

A variety of virulence factors, including “Yersiniabactin” class genes such as *fyuA*, *irp1*, *irp2*, *ybtA*, *ybtE*, *ybtP*, *ybtQ*, *ybtS*, *ybtT*, *ybtU*, and *ybtX*, were identified in 19 isolates (79.1%). These genes play a vital role in the pathogenicity of *K. pneumoniae* isolates. Moreover, 10 isolates (41.6%) harbored fimbrial adhesion determinants (e.g., *Ste*, *Stf*, and *Stb*), facilitating enhanced host cell adhesion, while one isolate (8.3%) contained *D-alanine-polyphosphoribitol ligase* and the streptococcal plasmin receptor/GAPD.

Other isolates exhibited extreme hypervirulence, attributed to their metabolic adaptations, including the presence of streptococcal enolase (*eno*), immune evasion through a polysaccharide capsule (similar to *gtaB*, *lytR*, and *manA*), stress adaptation via the manganese transport system (*mntB*), acid resistance mediated by urease (*ureB*), anaerobic respiration involving the nitrate reductase system (*narG* and *narH*), and cell surface integrity supported by the trehalose-recycling ABC transporter (*sugC*).

Collectively, these features allow for *K. pneumoniae* to survive in diverse host environments, evade immune responses, and exhibit heightened virulence, establishing it as a formidable pathogen associated with severe illnesses.

### 3.6. Plasmid Profiling

A diverse range of plasmid types conferring antimicrobial resistance was identified in the isolates (Figure 3). Most of these plasmids belong to the incompatibility (Inc) groups. Among the 24 clinical isolates analyzed, IncC plasmids were found in 83.3% of isolates, followed by IncR (79.1%) and IncL (8.3%). The 98 Kbps IncR plasmid, which harbors multiple genes responsible for fluoroquinolone and beta-lactam resistance in clinical *K.pneumoniae* isolates, was the most prevalent plasmid.

Seven variants of the IncF incompatibility group were identified: IncFIB(pQil) (91.6%), IncFIIK (79.1%), IncFII (41.6%), IncFIA(HI1) (37.5%), IncFIB(K) (16.6%), IncFIB(pKPHS1) (16.6%), and IncFIA (16.6%). Genomic analysis indicated that the IncF-type plasmid is approximately 101 Kbp in size. Plasmid maps revealed the presence of genes conferring resistance to aminoglycosides and carbapenems. Although incompatibility groups were identified, conjugation or transformation experiments were not conducted to confirm if the plasmids carry carbapenemase genes. Further functional studies are needed to verify.

Additionally, Col-type plasmids were detected, including Col(pHAD28) (12.5%), ColKP3 (66.6%), ColRNAI (54.2%), and Col4incI (29.1%). Among these, ColKP3 was the most observed Col-type plasmid and was determined to be 7605 base pairs in size. The genotype of the ColKP3 plasmid included the *bla_OXA-181_* gene (4141–4938), which encodes a class D β-lactamase enzyme capable of hydrolyzing β-lactam antibiotics, including carbapenems and extended-spectrum beta-lactamases (ESBLs). Although various plasmid types were associated with carbapenemase genes, no clear correlation was observed between specific incompatibility groups and the type of carbapenemase. Similarly, the distribution of virulence factors did not show a consistent pattern among carbapenemase-producing isolates. These findings underscore the genetic diversity and complexity of resistance plasmids and highlight the need for further large-scale studies to explore potential associations.

### 3.7. Mutation-Induced Resistance

The analysis of *K. pneumoniae* clinical isolates reveals several key mutations that contribute to antibiotic resistance by altering membrane permeability and enhancing efflux pump activity (Table 2). In terms of carbapenem resistance, mutations in porins like OmpK36 and OmpK37, including substitutions such as P170M, I128M, N230G, and A217S, were commonly observed. These mutations likely reduce drug influx, limiting the effectiveness of carbapenems. For cephalosporin resistance, additional porin mutations, like N495S, L50W, G619R, F198Y, and D223G, were identified, further impairing antibiotic entry and contributing to resistance. Efflux pump activity, which plays a significant role in fluoroquinolone resistance, was linked to mutations in the acrR gene, such as P161R, G164A, and P197L. These mutations lead to the overexpression of the AcrAB-TolC efflux system, enabling the bacteria to expel fluoroquinolones more efficiently. Additionally, mutations in the ramR gene, including P116L and P157* (a stop codon), further derepress the efflux pump, leading to increased resistance against both fluoroquinolones and tigecycline. Overall, the combination of porin mutations reducing membrane permeability and regulatory mutations enhancing efflux pump activity underpins the multidrug resistance observed in these K. pneumoniae isolates. This dual mechanism highlights the complexity of resistance strategies in these clinical strains.

### 3.8. Molecular Epidemiology of Carbapenem-Resistant K. pneumoniae (CRKP) in Pakistan

Using sequence type and regional representation as criteria, data extracted from pathogenwatch (a web-based platform developed by the center for genomic pathogen surveillance for real-time genomic surveillance of bacterial pathogens) included a total of 82 isolates: a total of 31 from ST147, 19 from ST231, 11 from ST29, and 21 from ST11 (Table 3). Genotypic analysis revealed the presence of numerous genes associated with antimicrobial resistance, particularly against beta-lactams and carbapenems. These include chromosomal genes (*bla_SHV_* and *bla_TEM_*) as well as acquired resistance gene variants (*bla_NDM_*, *bla_CTX-M_*, *bla_OXA_*, and *bla_DHA_*). The *bla*_NDM_ gene is a known carbapenemase gene. Some *bla*_OXA_ variants, like *bla*_OXA-48_ and *bla*_OXA-181_, also confer carbapenem resistance, but others encode narrow-spectrum or extended-spectrum beta-lactamases.

Among all isolates, *bla_NDM_* displayed five variants, while *bla_OXA_* had seven. The most common *bla_NDM_* variant in sequence type ST11 was *bla_NDM_*_-1_ (42.8%), followed by *bla_NDM-3_* (4.8%), *bla_NDM-6_* (4.8%), and *bla*_NDM-7_ (9.5%). Notably, *bla_NDM-1_* was present across all four sequence types (STs) but was most prevalent in ST11 (42.8%) and ST29 (45.45%). Meanwhile, *bla*_NDM-7_ was equally frequent in ST11 and ST147. Additionally, ST147 exhibited the highest incidence of *bla*_NDM-5_ (38.7%), followed by ST231 (20.8%).

Regarding *bla_OXA_*, five distinct forms were identified in ST11 and ST147. In ST11, the most common variant was *bla_OXA-1_* (61.9%), followed by *bla_OXA-48_* (14.4%), *bla_OXA-10_* (9.6%), *bla_OXA-181_* (4.8%), and *bla_OXA-232_* (4.8%). Conversely, ST147 had high prevalences of *bla_OXA-181_* (45.1%) and *bla_OXA-48_* (32.2%). Other *bla_OXA_* variants identified in ST147 included *bla_OXA-232_* (19.3%), *bla_OXA-505_* (9.7%), and *bla_OXA-10_* (3.2%). The *bla_OXA-232_* variant was predominantly found in ST231 (89.5%). In contrast, ST29 had a high prevalence of *bla_OXA-1_* (45.4%).

Another major class of acquired resistance genes present across all sequence types was *bla_CTX-M_*. For instance, *bla_CTX-M-15_* was identified in all four sequence types, with the highest prevalence in ST147 (96.7%), followed by ST11 (85.7%), ST231 (84.2%), and ST29 (72.7%). Other variants, including *bla_CTX-M-88_*, *bla_CTX-M-216_*_,_ and *bla_CTX-M-194_*, were found at negligible frequencies in their respective sequence types. For example, only one isolate from ST147 harbored *bla_CTX-M-88_* (3.2%), while *bla_CTX-M-216_* (9.0%) was found in ST29, and *bla_CTX-M-194_* (15.7%) was found in ST231.

The chromosomal resistance genes *bla_SHV_* and *bla_TEM_* were also observed. While *bla_TEM_* had two variants, *bla_SHV_* was represented by seven distinct forms. Appendix A illustrates the prevalence and distribution of beta-lactamase-producing organisms. Together, these resistance genes contribute to the production of the beta-lactamase enzyme, which deactivates the beta-lactam class of antibiotics.

A chi-square test of independence was performed to assess the association between sequence types (STs) and the presence of carbapenem resistance genes. The analysis revealed a statistically significant correlation between certain STs and specific resistance genes (*p* < 0.05). Notably, ST-147 showed a strong association with bla_NDM-5_ and bla_OXA-181_, while ST-231 was predominantly associated with bla_OXA-232_. ST-11 exhibited a high frequency of bla_NDM-1_ and bla_SHV-182_, and ST-29 had a distinct profile characterized by bla_NDM-1_ and bla_OXA-10_. These findings suggest that certain STs may serve as reservoirs for specific carbapenemase genes, indicating potential clonal dissemination of resistant strains within the population. This underscores the need for continuous molecular surveillance and targeted infection control measures (Appendix A).

### 3.9. Phylogenetic Analysis

The phylogenetic tree of *K. pneumoniae* illustrates the evolutionary relationship of core genomes among different isolates in comparison to a reference genome. Each branch represents a unique strain, with branch length indicating genetic distance. The shorter branches suggest high similarity to the reference, while the longer branches suggest significant divergence due to accumulated mutations. The isolate KP-289 was closely clustered with the reference genome, showing a high degree of similarity, while isolate KP-99 was also closely clustered with the reference genome, is likely to share a recent common ancestor, and may possess similar genetic traits, such as virulence or antibiotic resistance genes. The isolates KP-56 and KP-68 showed the most divergence from the reference genome. In the phylogenetic analysis, the reference genome acts as a baseline, and the tree provides a visual representation of how other strains have evolved from or diverged from the isolate from its reference genome. This comparative analysis can help to identify genetically related strains, track evolutionary patterns, and understand the molecular epidemiology of *K. pneumoniae* (Figure 4).

## 4. Discussion

Antimicrobial resistance is one of the most pressing challenges in global healthcare, as it compromises the efficacy of life-saving medications and complicates treatment protocols [21,27]. This study provides a detailed molecular epidemiological insight into carbapenem-resistant *K. pneumoniae* (CR-KP) in a clinical setting in Pakistan, contributing to the growing understanding of resistance dynamics in South Asia. *K. pneumoniae* is a major nosocomial pathogen, and its carbapenem-resistant strains are particularly alarming due to limited treatment options and high mortality rates associated with such infections [7,22]. Urine was the most commonly isolated specimen, consistent with studies from Pakistan and China [23,28], suggesting urinary tract infections as a primary clinical manifestation of CR-KP. A male predominance (64.7%) was observed, aligning with global studies showing similar male-to-female infection ratios [29,30]. A considerable proportion (37.5%) of isolates originated from critical care unit settings with high antimicrobial use and vulnerable patient populations. These findings underscore the need for strengthened infection prevention in ICU settings and a careful review of empirical antibiotic use policies, particularly regarding the use of agents such as carbapenems, colistin, cefiderocol, and novel beta-lactam–beta-lactamase inhibitor combinations, which are now considered last-resort options for treating multidrug-resistant Gram-negative infections [31,32,33]. This study revealed a carbapenem resistance rate of 42%, which is significantly higher than those reported from Ghana (5.6%) and Iran (24%) but somewhat comparable to China’s 34% [10,34,35,36], reinforcing the urgency for region-specific stewardship strategies.

Whole-genome sequencing (WGS) was used to characterize 24 CR-KP isolates. This technique allows for comprehensive resistance gene identification, high-resolution outbreak tracking, and targeted therapeutic interventions [37,38]. Resistance in *K. pneumoniae* can arise from chromosomal mutations or horizontal gene transfer. Mutations in genes like *gyrA* and *parC* confer fluoroquinolone resistance, while porin loss and efflux pump overexpression further reduce drug susceptibility [39,40,41,42]. These genetic alterations enable bacteria to survive under antibiotic pressure, leading to persistent infections and limited treatment outcomes.

This study identified a range of resistance determinants, including carbapenemase genes (*bla_NDM_*, *bla_OXA_*) and extended-spectrum beta-lactamase (ESBL) genes (*bla_CTX-M_*, *bla_TEM_*, *bla_SHV_*, *bla_CMY_*), which collectively contribute to multidrug resistance. The most prevalent variant was *bla_CTX-M-15_*, consistent with regional studies [43,44,45,46]. Among the carbapenemase genes, 84.61% of isolates carried a single resistance gene, while 46.18% had multiple resistance genes, illustrating a concerning trend of accumulating resistance determinants.

Molecular typing revealed four sequence types (STs): ST147 (53.8%), ST231 (30.76%), ST11 (7.7%), and ST29 (7.7%). Comparative analysis using Pathogenwatch showed that ST147 was the most frequently reported ST from Pakistan (34.78%), followed by ST11 (30.43%), ST231 (21.73%), and ST29 (14.49%). ST147 is globally recognized as a high-risk clone often linked to *blaNDM-1* and associated with nosocomial outbreaks [47,48]. Similarly, ST231, predominant in Oman and emerging in Pakistan, is increasingly detected in hospital and environmental settings [49,50]. ST11 is a globally disseminated clone, often carrying *blaKPC*, and has been implicated in outbreaks in China and other countries [10,51]. Our detection of ST29, previously found in environmental and veterinary settings in Pakistan, is the first to report this sequence type from clinical isolates in the country, suggesting a potential spillover from environmental reservoirs to human populations [52,53,54]. These findings emphasize the necessity for integrated surveillance under a One Health framework.

Genotypic analysis also highlighted the diverse *bla_NDM_* gene variants circulating in Pakistan. In our study, *bla_NDM-5_* (19.4%) and *bla_NDM-1_* (18.3%) were the most common, while *bla_NDM-7_*, *bla_NDM-3_*, and *bla_NDM-6_* were less frequent. Notably, ST11 was predominantly associated with *bla_NDM-1_* (42.8%), indicating its potential as a reservoir of high-level resistance. Previous data show that *bla_NDM_*-producing strains contribute significantly to both neonatal and adult mortality in the region [55,56]. This highlights the urgent need for targeted infection control and rational antibiotic use policies.

In addition to chromosomal mutations, plasmid-mediated resistance genes were also manifested. The frequent detection of plasmids across isolates reflects the high potential for horizontal gene transfer, accelerating the spread of resistance across hospitals and possibly into the community [57,58]. The identification of multiple high-risk STs, each carrying distinct resistance gene profiles, suggests the co-circulation of multiple successful clones with diverse evolutionary histories. This has serious implications for empirical therapy and infection control in hospital settings.

*K. pneumoniae* possesses various virulence factors that enhance survival and pathogenesis, including efflux pumps, siderophore systems, capsular regulators (*rmpA*, *RcsA*, *RcsB*), and phospholipase secretion systems [59,60,61]. The convergence of multidrug resistance and hypervirulence traits in circulating clones complicates treatment further and increases the risk of severe and hard-to-treat infections. Interestingly, while our isolates did not harbor the *bla_KPC_* gene, the detection of ST11 often associated with KPC production globally raises concerns about the future introduction and spread of KPC-producing strains in the region. The absence of *bla_KPC_* provides a valuable opportunity for early preventive measures. Continuous monitoring using WGS will be essential to detect its emergence and guide early containment efforts.

The epidemiological and molecular findings underscore the importance of integrating genomic data into national AMR surveillance frameworks. The identification of high-risk clones and gene variants should inform localized antimicrobial stewardship strategies, promote data sharing through platforms like Pathogenwatch, and prioritize rapid diagnostics in high-burden settings. Enhanced inter-hospital communication, strict infection control protocols, and regulation of antibiotic prescribing practices are essential to limit further dissemination.

## 5. Conclusions

Infections caused by carbapenem-resistant *Klebsiella pneumoniae* (CRKP) pose a significant threat to public health. One major concern is the spread of *Enterobacterales* strains carrying multiple drug resistance (MDR) genes. This study highlights the significant burden of carbapenem-resistant *Klebsiella pneumoniae* (CRKP) in Karachi, Pakistan. Resistance was mainly driven by the presence of *bla_OXA-232_*, *bla_NDM-5_*, and *bla_CTX-M-15_*, along with porin and efflux pump mutations that further limited antibiotic efficacy. High-risk sequence types, particularly ST147 and ST231, were commonly associated with multidrug resistance, emphasizing the potential for rapid dissemination. The findings underscore the urgent need for strengthened antibiotic stewardship, enhanced surveillance, and routine molecular diagnostics in clinical microbiology laboratories. However, this study’s limitations include a small number of sequenced isolates and a limited geographical scope. Future studies should incorporate a broader sample size, including multiple regions, and assess patient outcomes to better understand the clinical impact and guide effective infection control strategies. Robust nationwide surveillance will be essential for tracking the progression of resistance, shaping effective policies, and combating CRKP and other multidrug-resistant pathogens in Pakistan and beyond.

## Figures and Tables

**Figure 1 microorganisms-13-01577-f001:**
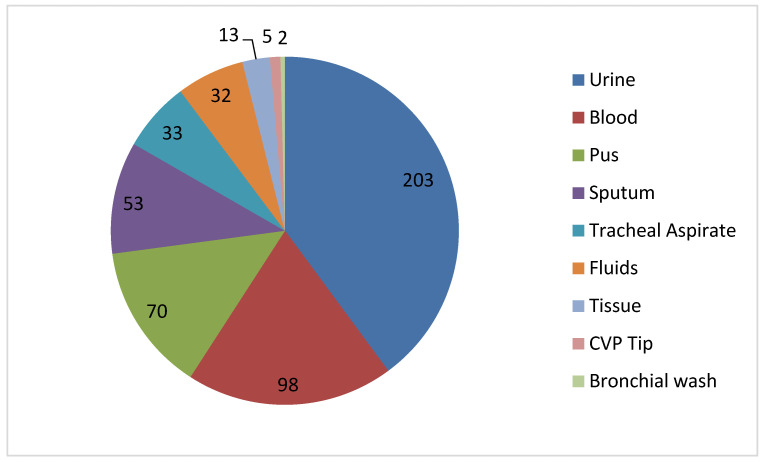
Distribution of clinical specimens from which *Klebsiella pneumoniae* was isolated, illustrating the frequency and diversity of sample sources across different clinical settings.

**Figure 2 microorganisms-13-01577-f002:**
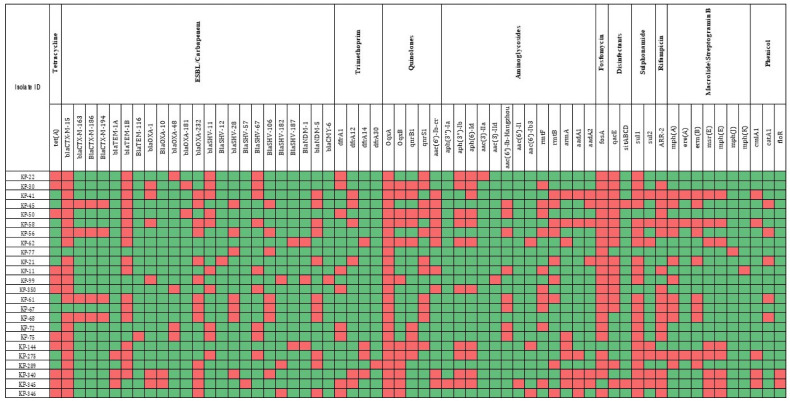
Heat map of antimicrobial resistance gene (Red = Gene Present, Green = Gene Absent).

**Figure 3 microorganisms-13-01577-f003:**
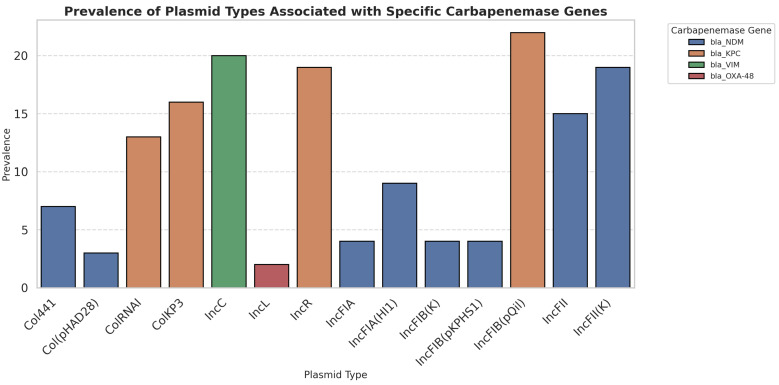
Frequency of plasmids detected in carbapenem-resistant *K. pneumoniae*.

**Figure 4 microorganisms-13-01577-f004:**
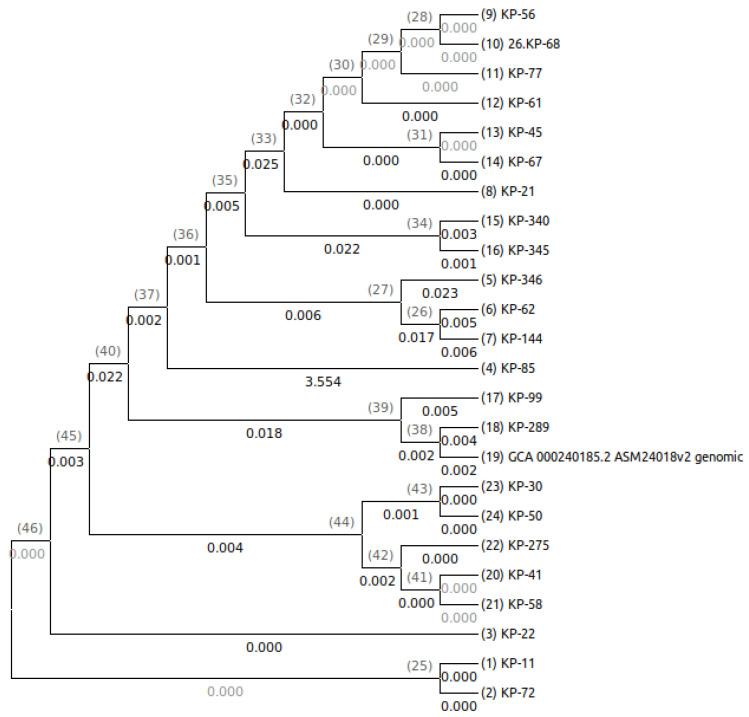
Carbapenem-resistant *K. pneumoniae* (CR-KP) isolates core genome-based evolutionary relationships.

**Table 1 microorganisms-13-01577-t001:** MIC of carbapenemase-producing CRKP.

S. No	Isolate ID	MIC of ANTIBIOTICS (μg/mL)															
		AK	CN	TOB	IMP	MEM	CAZ	FEP	PIP	TIC	TZP	TIM	LEV	CIP	SXT	ATM	CT
1	KP-22	2 S	1 S	1 S	16 R	16 R	64 R	64 R	128 R	128 R	128 R	128 R	8 R	4 R	320 R	64 R	2 I
2	KP-30	64 R	16 R	16 R	2 I	16 R	64 R	64 R	128 R	128 R	128 R	128 R	8 R	4 R	320 R	64 R	2 I
3	KP-41	64 R	16 R	16 R	16 R	16 R	64 R	64 R	128 R	128 R	128 R	128 R	8 R	4 R	320 R	64 R	2 I
4	KP-45	64 R	16 R	16 R	16 R	16 R	64 R	64 R	128 R	128 R	128 R	128 R	8 R	4 R	320 R	64 R	2 I
5	KP-50	64 R	16 R	16 R	2 I	16 R	64 R	64 R	128 R	128 R	128 R	128 R	8 R	4 R	320 R	64 R	2 I
6	KP-58	64 R	16 R	16 R	16 R	16 R	64 R	64 R	128 R	128 R	128 R	128 R	8 R	4 R	20 S	64 R	2 I
7	KP-56	64 R	16 R	16 R	16 R	16 R	64 R	64 R	128 R	128 R	128 R	128 R	8 R	4 R	320 R	64 R	2 I
8	KP-62	64 R	16 R	16 R	16 R	16 R	64 R	16 I	128 R	128 R	128 R	128 R	1 S	1 S	320 R	64 R	4 R
9	KP-77	64 R	16 R	16 R	16 R	16 R	64 R	64 R	128 R	128 R	128 R	128 R	8 R	4 R	20 S	64 R	2 I
10	KP-21	64 R	16 R	16 R	16 R	16 R	64 R	64 R	128 R	128 R	128 R	128 R	8 R	4 R	320 R	64 R	2 I
11	KP-11	64 R	16 R	16 R	16 R	16 R	64 R	64 R	128 R	128 R	128 R	128 R	8 R	4 R	320 R	64 R	2 I
12	KP-99	64 R	16 R	16 R	16 R	16 R	64 R	64 R	128 R	128 R	128 R	128 R	8 R	4 R	20 S	64 R	2 I
13	KP-350	64 R	16 R	16 R	16 R	16 R	64 R	64 R	128 R	128 R	128 R	128 R	8 R	4 R	320 R	64 R	2 I
14	KP-61	64 R	16 R	16 R	16 R	16 R	64 R	64 R	128 R	128 R	128 R	128 R	8 R	4 R	320 R	64 R	2 I
15	KP-67	64 R	16 R	16 R	16 R	16 R	64 R	64 R	128 R	128 R	128 R	128 R	8 R	4 R	320 R	64 R	2 I
16	KP-68	64 R	16 R	16 R	16 R	16 R	64 R	64 R	128 R	128 R	128 R	128 R	8 R	4 R	320 R	64 R	2 I
17	KP-72	64 R	16 R	16 R	16 R	16 R	64 R	64 R	128 R	128 R	128 R	128 R	8 R	4 R	320 R	64 R	2 I
18	KP-75	64 R	16 R	16 R	16 R	16 R	64 R	64 R	128 R	128 R	128 R	128 R	8 R	4 R	320 R	64 R	2 I
19	KP-144	64 R	16 R	16 R	16 R	16 R	64 R	64 R	128 R	128 R	128 R	128 R	8 R	4 R	320 R	64 R	2 I
20	KP-275	64 R	16 R	16 R	16 R	16 R	64 R	64 R	128 R	128 R	128 R	128 R	8 R	4 R	320 R	64 R	2 I
21	KP-289	64 R	16 R	16 R	16 R	16 R	64 R	64 R	128 R	128 R	128 R	128 R	8 R	4 R	320 R	64 R	2 I
22	KP-340	64 R	16 R	16 R	16 R	16 R	64 R	64 R	128 R	128 R	128 R	128 R	8 R	4 R	320 R	64 R	2 I
23	KP-345	64 R	16 R	16 R	16 R	16 R	64 R	64 R	128 R	128 R	128 R	128 R	8 R	4 R	320 R	64 R	2 I
24	KP-346	64 R	16 R	16 R	16 R	16 R	64 R	64 R	128 R	128 R	128 R	128 R	8 R	4 R	320 R	64 R	2 I

S: sensitive, R: resistant, AK: amikacin, CN: gentamicin, TOB: tobramycin, IMP: imipenem, MEM: meropenem, CAZ: ceftazidime, FEP: cefepime, PIP: piperacillin, TIC: ticarcillin, TZP: piperacillin–tazobactam, TIM: ticarcillin–clavulanate, LEV: levofloxacin, CIP: ciprofloxacin, SXT: trimethoprim-sulfamethoxazole, ATM: aztreonam, CT: colistin.

**Table 2 microorganisms-13-01577-t002:** Mutations that lead to antimicrobial resistance in *K. pneumoniae*.

Isolate	Mutations Change Membrane Permeability (Phenotypic Effect Against Carbapenem)	Mutations Change Membrane Permeability (Phenotypic Effect Against Cephalosporin)	Mutations Alter the Function of Efflux Pump (Phenotypic Effect Against Fluoroquinolones)	Transcriptional Repressor of Efflux Pump (Phenotypic Effect Against Fluoroquinolone and Tigecycline
KP-11	ompK37 p.I70M ompK37 p.I128M ompK37 p.N230G		acrR p.P161R acrR p.G164A acrR p.F172S acrR p.R173G acrR p.L195V acrR p.F197I acrR p.K201M	
KP-21	ompK37 p.I70M ompK37 p.I128M ompK37 p.N230G			
KP-22	ompK37 p.I70M, ompK37 p.I128M			
KP-30	ompK36 p.A217S, ompK37 p.I70M, ompK37 p.I128M	ompK36 p.N49S ompK36 p.L59V ompK36 p.G189T ompK36 p.F198Y ompK36 p.F207Y ompK36 p.T222L ompK36 p.D223G ompK36 p.E232R ompK36 p.N304E		
KP-41	ompK36 p.A217S, ompK37 p.I70M, ompK37 p.I128M	ompK36 p.N49S ompK36 p.L59V ompK36 p.G189T ompK36 p.F198Y ompK36 p.F207Y ompK36 p.T222L ompK36 p.D223G ompK36 p.E232R ompK36 p.N304E		
KP-45	ompK37 p.I70M ompK37 p.I128M ompK37 p.N230G			
KP-50	ompK36 p.A217S ompK37 p.I70M ompK37 p.I128M	ompK36 p.N49S		ramR p.L156V ramR p.E175D ramR p.F181*—Premature stop codon
KP-56	ompK37 p.I70M ompK37 p.I128M ompK37 p.N230G			
KP-58	ompK36 p.A217S ompK37 p.I70M ompK37 p.I128M	ompK36 p.N49S ompK36 p.L59V ompK36 p.G189T ompK36 p.F198Y ompK36 p.F207Y ompK36 p.T222L ompK36 p.D223G ompK36 p.E232R ompK36 p.N304E		
KP-61	ompK37 p.I70M ompK37 p.I128M ompK37 p.N230G			
KP-62	ompK36 p.A217S ompK36 p.N218H ompK37 p.I70M ompK37 p.I128M	ompK36 p.N49S ompK36 p.L59V ompK36 p.T86V ompK36 p.S89T ompK36 p.D91K ompK36 p.A93S ompK36 p.L191Q ompK36 p.F207W ompK36 p.Q227N ompK36 p.L229V ompK36 p.E232R ompK36 p.H235D ompK36 p.T254S		
KP-67	ompK37 p.I70M ompK37 p.I128M ompK37 p.N230G			
KP-68	ompK37 p.I70M ompK37 p.I128M ompK37 p.N230G			
KP-75	ompK37 p.I70M ompK37 p.I128M			
KP-77	ompK37 p.I70M ompK37 p.I128M ompK37 p.N230G			
KP-99	ompK36 p.A217S ompK37 p.I70M ompK37 p.I128M ompK37 p.N230G	ompK36 p.N49S ompK36 p.L59V ompK36 p.G189T ompK36 p.F198Y ompK36 p.F207Y ompK36 p.T222L ompK36 p.D223G ompK36 p.E232R ompK36 p.N304E		
KP-144	ompK36 p.A217S ompK36 p.N218H ompK37 p.I70M ompK37 p.I128M	ompK36 p.N49S ompK36 p.L59V ompK36 p.T86V ompK36 p.S89T ompK36 p.D91K ompK36 p.A93S ompK36 p.L191Q ompK36 p.F207W ompK36 p.Q227N ompK36 p.L229V ompK36 p.E232R ompK36 p.H235D ompK36 p.T254S		
KP-275	ompK36 p.A217S ompK37 p.I70M ompK37 p.I128M	ompK36 p.N49S ompK36 p.L59V ompK36 p.G189T ompK36 p.F198Y ompK36 p.F207Y ompK36 p.T222L ompK36 p.D223G ompK36 p.E232R ompK36 p.N304E		
KP-289	ompK36 p.A217S ompK37 p.I70M ompK37 p.I128M ompK37 p.N230G	ompK36 p.N49S ompK36 p.L59V ompK36 p.G189T ompK36 p.F198Y ompK36 p.F207Y ompK36 p.T222L ompK36 p.D223G ompK36 p.E232R ompK36 p.N304E		
KP-340	ompK36 p.A217S ompK36 p.N218H ompK37 p.I70M ompK37 p.I128M	ompK36 p.N49S ompK36 p.L59V ompK36 p.L191S ompK36 p.F207W ompK36 p.D224E ompK36 p.L228V ompK36 p.E232R ompK36 p.T254S		
KP-345	ompK37 p.I70M ompK37 p.I128M			
KP-346	ompK36 p.A217S ompK36 p.N218H ompK37 p.I70M ompK37 p.I128M	ompK36 p.N49S ompK36 p.L59V ompK36 p.L191S ompK36 p.F207W ompK36 p.D224E ompK36 p.L228V ompK36 p.E232R ompK36 p.T254S		

**Table 3 microorganisms-13-01577-t003:** Molecular epidemiology of carbapenem/beta-lactam resistance genes among different sequence types.

Genes	ST-147 (*n* = 31)	ST-231 (*n* = 19)	ST-29 (*n* = 11)	ST-11 (*n* = 21)	Total (*n*)	Percentage (%)
*blaCTX-M-15*	30	16	8	18	72	87.8
*blaCTX-M-88*	1	0	0	0	1	1.22
*blaCTX-M-194*	0	3	0	0	3	3.65
*blaCTX-M-216*	0	0	1	0	1	1.22
*blaDHA-1*	0	0	0	1	1	1.22
*blaDHA-7*	0	0	0	1	1	1.22
*blaNDM-1*	1	0	5	9	15	18.3
*blaNDM-5*	12	4	0	0	16	19.5
*blaNDM-7*	2	0	0	2	4	4.88
*blaNDM-6*	0	0	0	1	1	1.22
*blaNDM-3*	0	0	0	1	1	1.22
*blaOXA-1*	3	1	5	13	22	26.8
*blaOXA-10*	1	0	4	2	7	8.5
*blaOXA-181*	14	0	0	1	15	18.3
*blaOXA-232*	6	17	0	1	24	29.2
*blaOXA-48*	10	0	0	3	13	15.8
*blaOXA-505*	3	0	0	0	3	3.65
*blaOXA-9*	0	0	1	0	1	1.22
*blaSHV-100*	1	16	0	0	17	20.7
*blaSHV-67*	30	0	0	0	30	36.5
*blaSHV-12*	0	1	0	1	2	2.4
*blaSHV-155*	0	1	0	0	1	1.22
*blaSHV-66*	0	1	1	0	2	2.4
*blaSHV-182*	0	0	0	20	20	24.3
*blaSHV-187*	0	0	10	0	10	12.2
*blaTEM-1B*	27	16	9	5	57	69.5
*blaTEM-234*	0	0	1	0	1	1.22

## Data Availability

The Genomic data presented in this study are openly available in NCBI at https://www.ncbi.nlm.nih.gov/, reference number Bioproject PRJNA1012456.

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
