# Peer review of "The Microbiological Characteristics and Genomic Surveillance of Carbapenem-Resistant Klebsiella pneumoniae Isolated from Clinical Samples"

_microorganisms, 2025, doi:10.3390/microorganisms13071577_

Round 1

Reviewer 1 Report (Previous Reviewer 3)

Comments and Suggestions for Authors

Navigating the molecular epidemiology of carbapenem-resistant Klebsiella pneumoniae in Pakistan: a new lens on disease

The study reports molecular epidemiology of carbapenem-resistant K. pneumoniae in Pakistan. The paper has been improved compared to the original version but it still suffers from serious flaws which need to be addressed.

MAJOR COMMENTS

  1. There is no correlation between plasmid incompatibility groups and types of carbapenemases, it is also not clear if certain virulence factors are prevalent among specific carbapenemase producers
  2. In the results section, the genes encoding carbapenemases should be separated from those encoding ESBLs or narrow spectrum β-lactamases
  3. The quality of English language should be improved
  4. The introduction section should contain description of ESBLs, p-AmpC and narrow spectrum beta-lactamases encountered in K. pneumoniae since they rare listed in the Results section
  5. It should be clearly written in the abstract and results which carbapanemases are dominant among the isolates and which other beta-lactamases accompany them
  6. The title should be modified: there is no mention of diseases associated with analyzed pneumoniae isolates neither in abstract nor in Results section.

MINOR COMMENTS

  1. Abstract, line 27: MIC determination
  2. Abstract, line 31: colistin should be in lower case letters, the generic names of antibiotics should be in lower case
  3. Abstract, line 35: pneumoniae should be italicised, genus and species names of bacteria are written in cursiv
  4. Abstract, lines 37-45: bla is written in lower case letters
  5. Abstract, lines 37-45: the genes encoding carbapenemases should be presented separately from those encoding other beta-lactamases
  6. Page 3, line 100: what is the purpose of using chocolate agar, it is for Heamophilus and other fastidious bacteria
  7. Page 3, line 107: generic names of antibiotics are written in lower case letters
  8. Page 7, line 281: the sentence should be rephrased: contribute to resistance to carbapenems and expanded spectrum cephalosporins, extended-spectrum beta-lactamases are enyzmes, they are not antibiotics, the sentence in non-sense
  9. Page 6, Table 1, it is not clear how 24 isolates were selected for MIC determination? There were altogether 213 carbapenem-resistant isolates. Are these strains the same as those used for WGS? This should be clarified
  10. Page 6, line 259: Colistin is used for treatment of infections associated with carbapenem-resistant pathogens. We treat infection not the bacteria.
  11. Page 6, line 259: aztreonam is not afected by metallo β-lactamases, ESBL, p-AmpC and other carbapenemases hydroylze aztreonam. This should be clarified.
  12. Page 6, line 260: NDM should not be italicised
  13. Page 7, lines 262-265: generic names of antibiotics should be written in lower case letters
  14. Page 7, lines 279-289: the paper is focused on carbapenemases and thus the genes encoding these enzymes should be separated from those encoding ESBLs, p-AmpC and narrow spectrum beta-lactamases, blaTEM, blaCTX-M, blaOXA-1 and blaCMY genes have nothing to do with carbapenem resistance.
  15. Page 10, line 30: incL
  16. Page 15, Fig. 3: The plasmid incompatibility groups should be presented according to the type of carbapenemase, It is well known that some plasmid types carry certain carbapenemase genes, for instance IncA/C usually carried VIM encoding genes whereas IncL frequently harbours blaOXA-48
  17. Table 2 is unreadible, the font size should be increased
  18. Page 16, line 98: pneumoniae
  19. Conclusions, line 195: we do not effectively treat MDR bacteria, we treat infections caused by MDR bacteria
  20. References are not arranged according to the MDPI style
Comments on the Quality of English Language

The quality of English language should be improved. Some parts of the text are difficult to understand. The style or writting should be improved. 

Author Response

Reviewer 2 Report (New Reviewer)

Comments and Suggestions for Authors

See the attachement

Comments on the Quality of English Language

The quality of Engligh language in this manuscript should be improved by a native speaker.

Author Response

Reviewer 3 Report (New Reviewer)

Comments and Suggestions for Authors

This manuscript, entitled “Navigating the Molecular Epidemiology of Carbapenem-Resistant Klebsiella pneumoniae in Pakistan: A New Lens on Disease.” had described the phenotypic and genomic characterization of K. pneumoniae isolated from various clinical sources in Karachi, Pakistan. This study has potential interest, however, it lacks novelty and has limited results provided.

The conclusion yield from this study was regular outcomes without much novel and significant finding. What’s the highlight of this study?

213 strains is too few and the strain collection period (July 2022 to June 2023) is too short for a surveillance study. I suggest the authors either expending the isolation time providing more information regarding the CRKP. For example, digging into the genomes and compare them with other studies. Either would be helpful to add up the importance and novelty.

The data are poorly organized and presented.

Author Response

Reviewer 4 Report (New Reviewer)

Comments and Suggestions for Authors

The present study investigates the phenotypic and genomic characterisation of K. pneumoniae isolated from various clinical sources in Pakistan. The aim is to shed light on the current state of antibiotic resistance among K. pneumoniae isolates in Pakistan through a thorough evaluation of both phenotype and genotype. The findings obtained by the authors align with the methodologies employed. The available data on the resistance of this micro-organism in the country are reported and discussed. I have no criticism, except that the abstract should be reduced by reporting the essential data to improve readability.

Round 2

Reviewer 1 Report (Previous Reviewer 3)

Comments and Suggestions for Authors

MAJOR COMMENTS

The paper needs extensive language and style editiing. In the present form it is difficult to understand.

The authors lack basic microbiology knowledge, for instance CMY-16 is not narrow spectrum beta-lactamases, it is p-AmpC, narrow spectrum are TEM-1 and TEM-2 and SHV-1. They hydrolyze only penicillins. They do not know that OXA-181 can cleave beta-lactam antibiotic, certainly not sn ESBL, another beta-lactam hydrolyzing enzyme. Some sentences are complete nonsense. Enzymes hydrolyze antibiotics and not other enzymes.

MINOR COMMENTS

  1. Abstract, line 21: Klebsiella pneumoniae-italicised, the bacterial name should be written in full when mentioned for the first time, later abbreviation is used
  2. Abstract, line 27-30: bla should be italicised
  3. Abstract, line 31: followed by instead accompanied by
  4. Introduction, line 48-use abbreviation CRKP for carbapenem-resistant Klebsiella pneumoniae
  5. Material and methods, line 104: what is the purpose for using chocolate agar for Enterobacterales? It should be used for Haemophilus
  6. Materal and methods, line 109: The subheading title should be: Antimicrobial susceptibility testing, omit to check......
  7. Material and methods, line 111: what is different antibiotics of carbapenem, I assume of carbapenem group? Does it mean imipenem, meropenem and ertapenem?
  8. Materal and methods, line 116: spaces is missing between 19 and mm
  9. Material and methods, line 123: The title of subheading should be phenotypic detection of carbapenemases, omit in CRKP, it is written in the title that CRKP are analyzed
  10. Material and methods, line 129: define what is high MIC
  11. Material and methods, line 168: how was MLST done? Did the authors perform PCR for seven housekeeping genes or STs were retrieved from WGS? It is not clear
  12. Results, line 211: isolates-lower case, collected instead of submitted
  13. Results, line 237: The title of the subheading should be: Antibiotic susceptibility, it is not necessary to explain every time that CRKP are analyzed
  14. Results, line 244: including imipenem and meropenem instead of especially
  15. Results, line 246: 128 µg/ml is MIC value and not resistance level
  16. Results, line 254-256: the results should be presented in past tense and not in present because this is already done
  17. Results, line 258: of this aminoglycoside
  18. Results, line 259: what is moderate resistance? The strains are either resistant, intermediate susceptible or susceptible. There is no category moderate resistant
  19. Results, line 274: The title of the Table should be: MICs of CRKP, omit CarbaNP test
  20. Results, page 8, line 308: blaNDM-bla italicised, NDM subscript
  21. Results, page 8, line 310: blaCMY-16 is p-AmpC gene and not gene encoding narrow spectrum beta-lactamase, the authors should check if the isolate is resistant to cefoxitin which is in line with AmpC beta-lactamases
  22. Results, Virulence factors: there is no link between virulence factors and resistance genes and clinical presentation of infection. It is not clear if some virulence traits are associated with severe infections such as sepsis or meningitis.
  23. Results, plasmid profiling: Identification of certain plasmid incompatibility groups is not proof that these plasmids carry resistance genes. The authors should do the conjugation or transformation experiments and characterize plasmids in tranconjugants or transformants in order to make sure that certain plasmid types carry carbapenemase encoding genes.
  24. Results, plasmid profiling, line 42: an enzyme can hydrolyzing beta-lactam antibiotic, certainly not ESBL which is an enzyme as well, the sentence is complete nonsense
  25. Page 12, line 2: pneumoniae should be abbreviated as it is mentioned already many time in the text
  26. Page 14, line 30: blaOXA genes are not all carbapenemase encoding genes. Many of them encode narrow spectrum beta. Lactamases (OXA-1 to OXA-4) or ESBLs.
  27. Page 17, line 102: carbapenems are no longer last resort antibiotics for infections due to MDR gram-negative bacteria. Now, colistin, cefiderocol and novel beta-lactam-inhibitor combinations are the last resort antibiotics.

Author Response

Reviewer 2 Report (New Reviewer)

Comments and Suggestions for Authors

Comments on the Quality of English Language

A native-speaker should correct the manuscript.

Author Response

Reviewer 3 Report (New Reviewer)

Comments and Suggestions for Authors

I appreciate the response from the authors. However, the major issue of this study remain. Although the study had performed the analysis of the phenotypic and genomic characteristics of Klebsiella pneumoniae in Karachi, Pakistan. It doesn’t mean this study has novelty and significance. Although such data remains scarce in the region, it doesn’t mean it has significance. This region is too specific. Please try to answer the following questions. Firstly, what novel point or finding does this study have? Secondly, are there similar data in other regions? What’s the similarity and differences?

Author Response

This manuscript is a resubmission of an earlier submission. The following is a list of the peer review reports and author responses from that submission.

Round 1

Reviewer 1 Report

Comments and Suggestions for Authors

The manuscript submitted by Rizvi et al. aimed to perform a detailed genomic characterization of 13 isolates of carbapenem-resistant K. pneumoniae. Despite the small number of isolates, initially, 6125 isolates of enterobacteria were evaluated, resulting in 508 isolates of K. pneumoniae, of which only 13 were considered CR-KP. However, these 13 isolates were assessed for their genetic resistance profile, STs, plasmids, and virulence, demonstrating the authors' commitment to conducting a well-executed genomic characterization using NGS.

I suggest that the manuscript undergo English language revision to adhere to the standards of formal writing. While the English is understandable, the writing and subsequent translation into English hinder the comprehension of some passages and compromise the overall quality of the manuscript. I recommend considering having the manuscript revised by a native English speaker. For example, see line 169: "the above said enzyme."

Remarks:

In the keywords, avoid repeating words already present in the title. Use other terms that contextualize the study area for better article indexing.

Line 16 - bacterium-Klebsiella pneumoniae - why this sentence structure?

Line 38 – Check the grammatical structure of the sentence; it is incorrect.

Line 46 – B-lactamase is the enzyme, not the group of antibiotics.

Line 49 – Abbreviate the genus. The genus should be written out in full only the first time it appears in the manuscript. Subsequently, use the abbreviation. Double-check this usage (line 85, 92, 93 contains the same error, among other lines).

Line 59 – Follow the journal's reference format.

Line 65 “in connection to above” – improve this construction.

Include a reference for section 2.2.

Line 89 – Include the version/year of CLSI considered.

 Line 99 – Reference the extraction method or describe it.

Lines 143-145 – I don't understand why the authors describe the STs of the study isolates in this section. Review and do not anticipate your results.

Improve the titles of the tables (such as Table 1) to be self-explanatory.

In Table 1, the authors include the results of various antibiotics not described in the methodology. Adjust to allow understanding of what was actually evaluated in the study.

I don't understand why the authors included tables 1 and 2. Do they show the same results? I believe you can keep table 2 and include the information from table 1 in text form in the manuscript.

 Sections 3.4, 3.5, and others: Use italics for all gene names in the manuscript. Double-check.

In the discussion, I suggest that the authors explore more on the epidemiology of KPC dissemination and possible strategies to combat its spread. Show how the results of the present research can contribute to a better understanding and control of this serious public health problem.

Line 206 of the conclusion – italicize all scientific names. Double-check in the manuscript.

The conclusion should be reformulated. It is presented in a very extensive and long manner, consisting of a repetition of results. Be concise and direct in concluding the study. What do the obtained data actually allow to conclude?

Comments on the Quality of English Language

I suggest that the manuscript undergo English language revision to adhere to the standards of formal writing. While the English is understandable, the writing and subsequent translation into English hinder the comprehension of some passages and compromise the overall quality of the manuscript. I recommend considering having the manuscript revised by a native English speaker. For example, see line 169: "the above said enzyme."

Author Response

Point to point response has been attached

Reviewer 2 Report

Comments and Suggestions for Authors

In this manuscript the authors aimed at describing the molecular characteristics of 508 clinical isolates of K. pneumoniae, focusing on carbapenem-resistant strains.

The topic is interesting, but there are some important flaws in the design of the study and on the execution of the analyses.

The most important issue is related to the antimicrobial susceptibility analysis reported in table 1. According to the data presented, 41.9% of the isolates tested resistant to meropenem, but only 1.1% were resistant to ampicillin alone. This is quite astonishing and raise some doubts regarding the accuracy of the antimicrobial susceptibility testing. The authors should provide an explanation on these results and on the potential underlying mechanisms.

Moreover, only 13 isolates were analyzed through whole genome sequencing, and we cannot be sure that their mechanisms of resistance are representative of the entire series. Finally, limited information is provided regarding some important determinants of resistance against carbapenems, such as porin mutation and consequent reduced permeability.

Comments on the Quality of English Language

The English language requires extensive revision. 

Author Response

(The authors gave the same response as above.)

Reviewer 3 Report

Comments and Suggestions for Authors

MICROORGANISMS

The study analyzed phenotypic and genomic characteristics of K. pneumoniae from hospitals in Pakistan. The study found the isolates to harbour OXA-48, OXA-181 and NDM carbapenemases alongside with ESBLs belonging to CTX-M family. The isolates belonged to ST147, ST231, ST29 and ST11. Different types of plasmids were found: IncFIB, IncFII ect.

MAJOR COMMENTS

1.      The manuscript is written in a very confusing way and is difficult to follow and comprehend

2.      Phenotypic testing for ESBL and p-ampC was not done. It is important to determine all resistance traits to beta-lactam antibiotics because the genes encoding ESBLs or p-AmpC often reside on the same plasmid carrying carbapenemases.

3.      The authors lack basic microbiology knowledge. Carbapenems belong to β-lactam family. β-lactam antibiotics are classifed as penicillins, cephalosporins, monobactam and carbapenems. The authors state that they analysed molecular epidemiology of β-lactam and carbapenem resistance genes. This is nonsense sentence. Moreover, they tested the susceptibility of K. pneumoniae to ampicillin. There is intrinsic resistance of K. pneumoniae to penicillins due to instrinsic, chromosomally encoded SHV-1 or SHV-11 β-lactamase. 

4.      The panel for antibiotic susceptibility is incomplete.  Antibiotics which exert excellent activity against CRE are not included in the panel for testing, for example novel combinations with inhibitors and cefiderocol.

5.      There is no link between carbapenemase production and plasmid types. In order to characterize plasmids it is necessary to perform conjugation or transformation and to determine resistance patterns and resistance genes content of the transconjugants and the transformants. Iti s not clear which plasmids encoded resistance traits. It is well known that IncL plasmid carries OXA-48 genes whereas IncX or IncA/C harbours NDM encoding genes.  

6.      There is no link between carbapenemase production and virulence traits. It is well known that hypervirulence is often associated with NDM or OXA-48 carbapenemases.

7.      Thirteen strains is too small number of isolates to make conclusions. It would be necessary to analyze at least 30 to 40 isolates. Resistance genes and plasmid characterization can be done without WGS which is expensive.

8.      The quality of English language is insuficient which makes the text difficult to  understand. The paper should be proofread by a native English speaker. There are a lot of grammatical and typographical errors throughout the text. Extensive editing is necessary.

MINOR COMMENTS

1.      bla should be italicised and the name of beta-lactamase gene written is subscript.

2.      Line 21: The isolates were tested for resistance to carbapenems and presence of carbapenemases. Iti s not necessary to explain that carbapenemases are enzymes.

3.      Line 27: The molecular epidemiology of beta-lactam resistance genes was demonstrated. The sentence should be rephrased. The genes do not exhibit resistance. The resistance is mediated by proteins which are products of the genes.  

4.      Line 39: virulence factors (plural)

5.      Line 72-74: the sentence should be rephrased. I do not understand what the authors meant

6.      Line 172: 13 K. pneumoniae were subjected instead of exposed to WGS

7.      Table 1: Iti s really strange that only 1% of the isolates is resistant to ampicillin in light of the fact that there is intrinsic resistance to penicillins in K. pneumonae. Even more unexpected is the rate of colistin resistance of 82%. If this is true there is nothing left to treat the patients. Under antibiotic class there is penicillin/cephalosporin combination. Which combination was tested? To my knowledge penicillins and not combined with cephalosporins because such combinatins exert antagonism. Penicillins can be combined with aminoglycosides, but not with cephalosporins.

8.      Line 93: Klebsiella pneumoniae with first letter in upper case.

9.      The abstract should cleary demonstrate the rate of certain types of carbapenemases, for instance OXA-48, OXA-181 and NDM.

10.  Line 25. Based on sequence types the molecular epidemiology….The sentence should be rephrased. The first part of the sentence is not linked to the second part. The sentence is nonsense.

Comments on the Quality of English Language

The quality of English language is not sufficient. There are a lot of grammatical and typographical errors which make the manuscript difficult to understand. The paper should be proofread by a native English speaker. 

Author Response

(The authors gave the same response as above.)
